# Phase-stabilised self-injection-locked microcomb

**Thibault Wildi[1,3], Alexander E. Ulanov[1,3], Thibault Voumard[1], Bastian Ruhnke[1] & Tobias Herr[1,2]** ✉

Microresonator frequency combs (microcombs) hold great potential for precision metrology within a compact form factor, impacting a wide range of applications such as point-of-care diagnostics, environmental monitoring, time-keeping, navigation and astronomy. Through the principle of self-injection locking, electrically-driven chip-based microcombs with minimal complexity are now feasible. However, phase-stabilisation of such self-injection-locked microcombs—a prerequisite for metrological frequency combs—has not yet been attained. Here, we address this critical need by demonstrating full phase-stabilisation of a self-injection-locked microcomb. The microresonator is implemented in a silicon nitride photonic chip, and by controlling a pump laser diode and a microheater with low voltage signals (less than 1.57 V), we achieve independent control of the comb's offset and repetition rate frequencies. Both actuators reach a bandwidth of over 100 kHz, enabling phase-locking of the microcomb to external frequency references. These results establish photonic chip-based, self-injection-locked microcombs as low-complexity yet versatile sources for coherent precision metrology in emerging applications.

Optical frequency combs provide large sets of laser lines that are equidistant in optical frequency and mutually phase-coherent[1,2]. Owing to this property, they have enabled some of the most precise measurements in physics and are pivotal to a vast range of emerging applications, from molecular sensing to geonavigation. Frequency combs based on high-Q nonlinear optical microresonators (microcombs)[3,4] that can be fabricated in complementary metal-oxide-semiconductor (CMOS) compatible, low-cost, scalable, wafer-scale processes[5,6], promise to bring frequency comb technology into widespread application beyond the confines of optics laboratories[7–9].

In microcombs, nonlinear processes partially convert a continuous-wave (CW) driving laser with frequency $\nu_p$ into a series of comb lines that are mutually spaced in frequency by the comb's repetition rate $f_{rep}$, so that $\nu_\mu = \nu_p + \mu f_{rep}$, describes the frequencies $\nu_\mu$ in the comb ($\mu = 0, \pm 1$, is a mode index relative to the pump; see Fig. 1c). For many comb-based precision measurements, it is crucial to independently control the comb's defining parameters, here $\nu_p$ and $f_{rep}$, on

a level that permits full phase control, i.e. *phase-locking*, of $\nu_p$ and $f_{rep}$ to external frequency references. This is equivalent to controlling carrier wave and envelope of the temporal optical waveform as indicated in Fig. 1d. For instance, this is important for molecular spectroscopy, environmental monitoring, medical diagnostics, geonavigation, exoplanet search, and other emerging applications that rely on phase-coherent links between electromagnetic waves.

A major advancement in microcombs came through the principle of self-injection locking (SIL)[10–12], which enabled electrically-driven comb sources with drastically reduced operational complexity and chip-level integration[13–20]. Instead of a low-noise tabletop pump laser, SIL utilises a chip-scale semiconductor pump laser and a narrow linewidth injection feedback from a high-Q microresonator. The SIL mechanism leads to a low-noise pump laser and elegantly ensures that the laser is intrinsically tuned to the microresonator for comb generation. Although highly attractive, the simplicity and compactness of SIL-based combs entail a critical drawback with regard to controlling $\nu_p$

---

[1]Deutsches Elektronen-Synchrotron DESY, Notkestr. 85, 22607 Hamburg, Germany. [2]Physics Department, Universität Hamburg UHH, Luruper Chaussee 149, 22761 Hamburg, Germany. [3]These authors contributed equally: Thibault Wildi, Alexander E. Ulanov. ✉e-mail: tobias.herr@desy.de

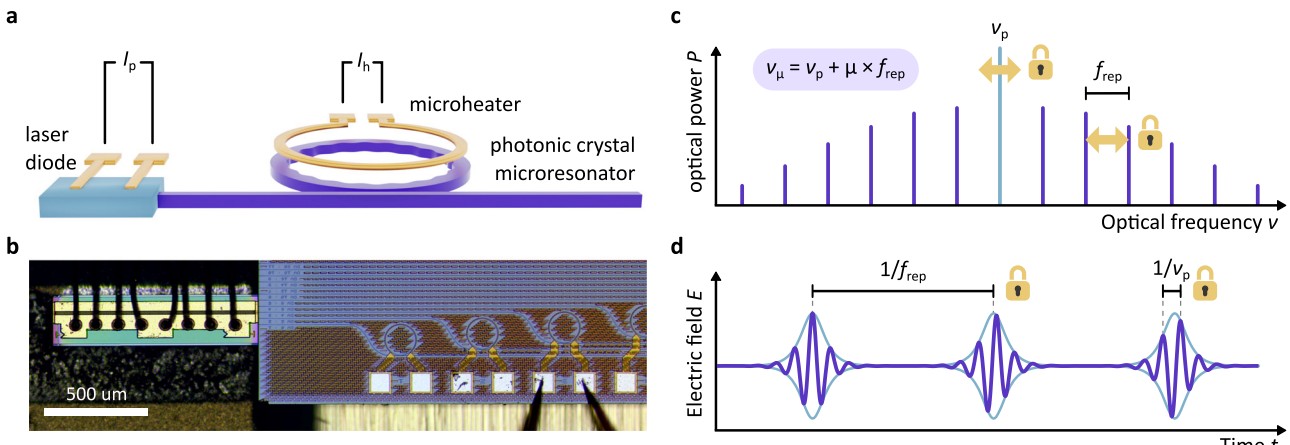

**Fig. 1 | Phase-stabilised self-injection-locked microcomb. a** Synthetic-reflection self-injection-locked microcomb. The combined actuation of the laser diode current $I_p$ and the current $I_h$ of a microheater controlling the microring temperature enables full phase-stabilisation of the microcomb via low voltage signals. **b** Micrograph of the SIL microcomb source comprised of a DFB laser diode (left) butt-coupled to a photonic-chip hosting $Si_3N_4$ microresonators (right). A metallic microheater embedded in the $SiO_2$ cladding is routed above the microring. **c** The optical spectrum of a continuous-wave driven microcomb is comprised of equidistant lines $v_\mu$ spaced by the comb's repetition rate $f_{rep}$ and centred on the pump frequency $v_p$. Full phase-stabilisation of the microcomb entails locking both degrees of freedom to an external reference. **d** In the time domain, this corresponds to a pulse train with a stabilised period $\tau_{rep} = 1/f_{rep}$ and optical carrier period $\tau_p = 1/v_p$.

and $f_{rep}$. In contrast to previous non-SIL systems in which the frequency and the power of a tabletop pump laser have been used as independent actuators to simultaneously phase-stabilise $v_p$ and $f_{rep}$[21–23], in SIL systems, these parameters are not independent (both depend on the laser pump current). Previous work has already accomplished stabilisation of one degree of freedom ($f_{rep}$)[24], however, phase-stabilisation of both degrees of freedom is an outstanding challenge. This lack of full phase-stabilisation in SIL microcombs represents a serious shortcoming for metrological applications.

Here, we present a chip-scale, electrically-driven, metrology-grade SIL microcomb operating at CMOS-compatible voltages. This source combines a semiconductor laser diode and a high-quality factor silicon nitride microresonator equipped with an integrated microheater[25,26] in a compact millimetre-square footprint (see Fig. 1a, b). The diode current and the integrated microheater provide two independent, low-voltage (<1.5 V) actuators reaching a remarkable >100 kHz effective actuation bandwidth. With these actuators, and in conjunction with synthetic reflection SIL[20], which supports a large range of laser detunings and lowers the actuation bandwidth requirement through laser linewidth narrowing, we demonstrate full phase stabilisation of the microcomb by phase-locking $v_p$ and $f_{rep}$ to external frequency references, creating a small-footprint, low-complexity, low-cost and CMOS-compatible frequency comb for demanding metrological applications.

## Results
### Setup
Our microcomb is based on CW laser-driven dissipative Kerr-solitons (DKS)[4,7,27] in a chip-integrated silicon nitride photonic crystal ring resonator (PhCR)[20,28–30]. In this scheme, a semiconductor distributed feedback (DFB) laser diode is butt-coupled to the photonic chip hosting the PhCR (coupling losses of ~3.5 dB), delivering approximately 25 mW of on-chip optical pump power at 1557 nm; a cleaved ultra-high numerical aperture optical fibre (UHNA-7) is utilised for output coupling (coupling losses of ~1.7dB). Both the laser chip and microresonator chip are temperature stabilised with a precision of ±5 mK via standard electric heaters/coolers. The microresonator itself is characterised by a free-spectral range (FSR) of 300 GHz, anomalous group velocity dispersion, and a high quality factor $Q \approx 1.5 \times 10^6$ (see Methods). An integrated metallic microheater[25,26] is embedded in the

silica cladding above the resonator waveguide for fast thermal actuation of the microresonator. Complementary to piezo-electric or electro-optic actuators[31–34], which in an integrated setting can also reach high actuation bandwidth, microheaters are an attractive low-complexity alternative as they provide a robust, long-lifetime and low-voltage solution, that is readily compatible with silicon-based photonic chip technology.

By leveraging a recently demonstrated synthetic reflection technique[20], where the nano-patterned corrugation of the PhCR generates a tailored optical feedback, robust self-injection locking of the driving laser diode is achieved. This also has the desirable effect of forcing exclusive and deterministic single-soliton operation[20,28]. Synthetic reflection also substantially extends the range of pump frequency-to-resonance detunings that are permissible during comb operation, providing extended actuation range and robust operation under phase-locking conditions. Moreover, the laser linewidth narrowing obtained via SIL relaxes the need for high bandwidth actuation.

In SIL DKS operation, the DFB laser's emission frequency $v_p$ (the central comb line of the microcomb) can be tuned by adjusting the current around the set point at a rate of 27 MHz mA$^{-1}$, which also affects the DKS repetition rate by 160 kHz mA$^{-1}$ via the detuning-dependent Raman-induced soliton self-frequency shift (see Supplementary Information). A second degree of freedom is provided by the microheater, which we operate at a current bias of 3 mA (105 mV). Via the microheater, the microcomb's repetition rate $f_{rep}$ can be tuned with a sensitivity of ~400 kHz mA$^{-1}$. As the laser diode and the microresonator are coupled through SIL, the microheater also induces a shift in the microcomb's centre frequency $v_p$ (pump line) with a sensitivity of ~160 MHz mA$^{-1}$. The actuator sensitivities are summarised in Table 1 and a theoretical derivation is provided in

## Table 1 | Actuator sensitivities

| Sensitivity | $f_{off}$ | $f_{rep}$ |
|---|---|---|
| $I_p$ | 27 MHz/mA | 160 kHz/mA |
| $I_h$ | 160 MHz/mA | 400 kHz/mA |

Sensitivity of the SIL microcomb's offset frequency $f_{off}$ and repetition rate $f_{rep}$ to the DFB current $I_p$ and micro-heater current $I_h$. The values were measured around the experiment set point of ~180 mA and 3 mA, respectively. A theoretical derivation of the actuator tuning coefficients is presented in Section 1 and 2 of the Supplementary Information.

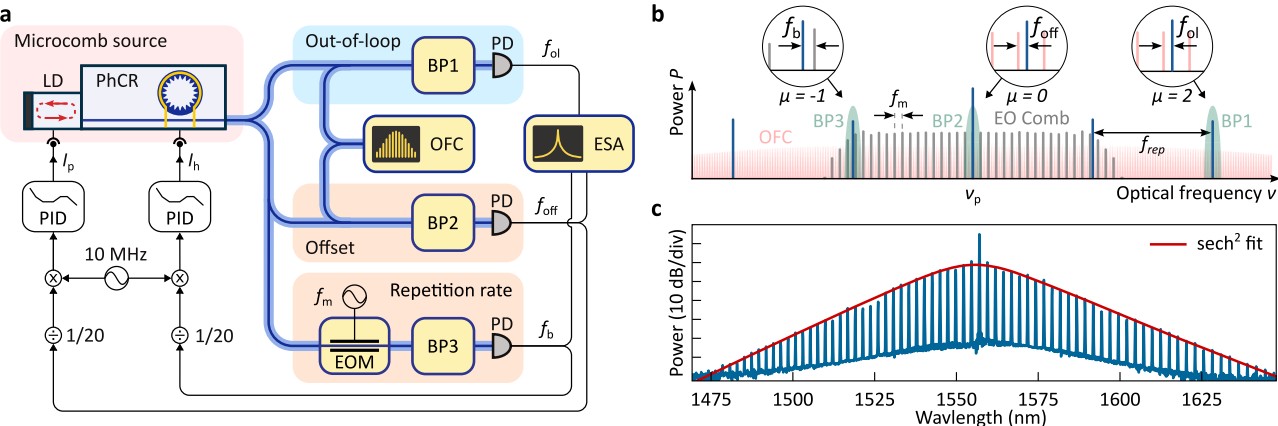

**Fig. 2 | Experimental setup. a** The microcomb source, a laser diode self-injection-locked to a photonic crystal ring resonator (PhCR) operating in the dissipative Kerr-soliton regime, is stabilised via two phase-lock loops feedback controlling the diode current $I_p$ and heater current $I_h$ respectively. OFC: reference optical frequency comb; ESA: electronic spectrum analyser; EOM: electro-optic modulator; BP: band-pass filter; PD: photodetector; LD: laser diode; PhCR: photonic crystal ring resonator; PID: proportional-integral-derivative controller. **b** Frequency diagram, depicting the self-injection-locked microcomb (blue), the reference 1 GHz oscillator (red) and the electro-optic (EO) modulation comb (grey). The frequencies $f_b$, $f_{off}$, and $f_{ol}$, corresponding to the repetition rate, offset, and out-of-loop beat notes, respectively, are extracted by the optical band-pass filters BP1-3 (green) as shown in the insets. **c** Optical spectrum of a self-injection-locked microcomb. The spectrum is well fitted by a sech$^2$ envelope with a FWHM of 1.44 THz.

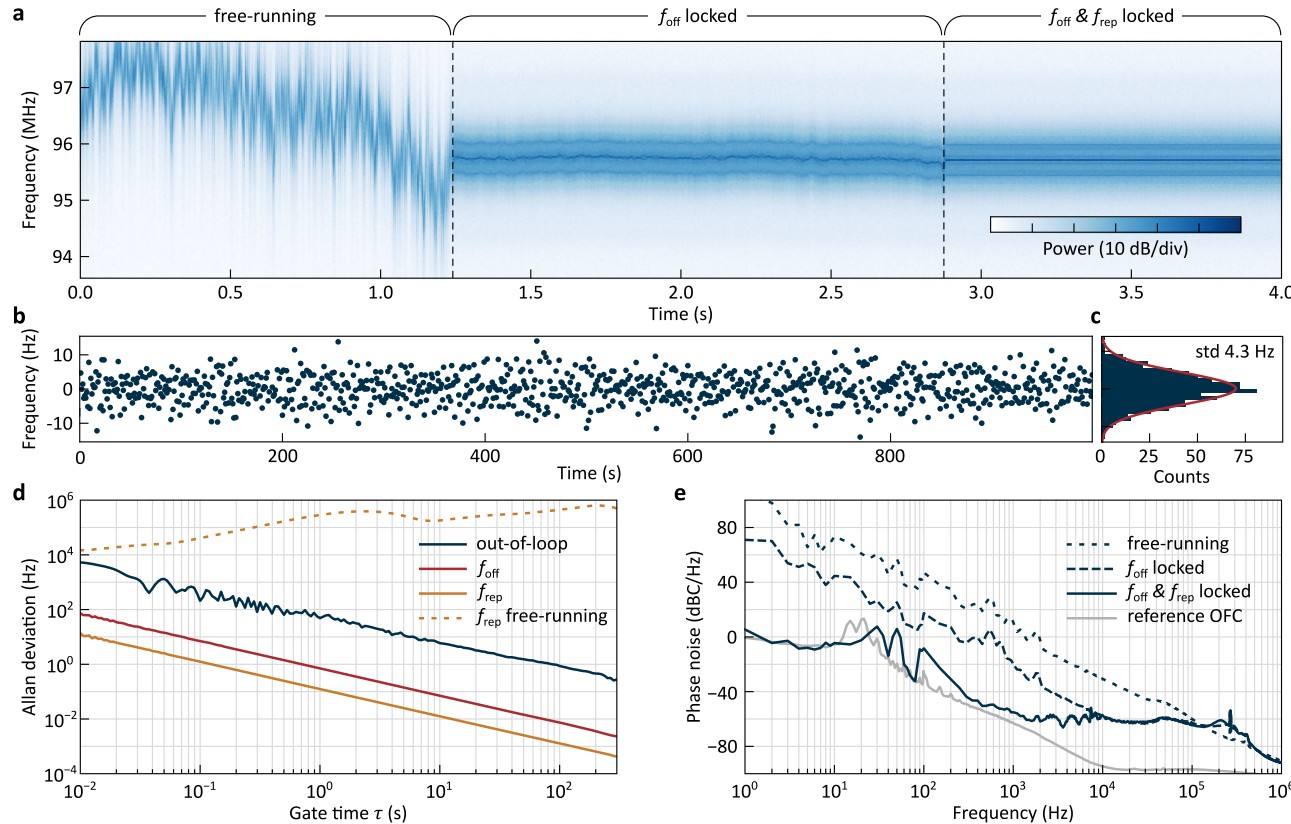

**Fig. 3 | Full phase-stabilisation of the self-injection-locked microcomb.**
**a** Spectrogram of the out-of-loop beat note $f_{ol}$ showing the transition from a free-running to a fully-locked state through the successive initiation of the offset $f_{off}$ and repetition rate $f_{rep}$ locks. **b** Time series measurement of the out-of-loop frequency $f_{ol}$ in the fully-locked state. The samples are acquired using a 1 s gate time at a rate of 1 Hz (zero dead time). **c** Histogram of the values shown in (**b**), with a Gaussian fit (in red) with 4.3 Hz standard deviation (std). **d** Overlapping Allan deviations (OADs) under full-phase stabilisation of the out-of-loop signal (solid, blue), the in-loop offset frequency $f_{off}$ (solid, red) and microcomb repetition rate $f_{rep}$ (solid, orange) as a function of the gate time $\tau$. The OADs average down with $\tau^{-0.996}$, $\tau^{-0.997}$ and $\tau^{-0.922}$ respectively. The OAD of the free-running $f_{rep}$ is provided for comparison (dashed, orange). The frequency counter noise floor, 40dB below the level of $f_{rep}$, is not shown. **e** Single-sideband phase noise of the out-of-loop beat note $f_{ol}$ in the free-running (orange), offset-locked (red) and fully-locked states (blue). The phase noise of the reference optical frequency comb is also shown (grey).

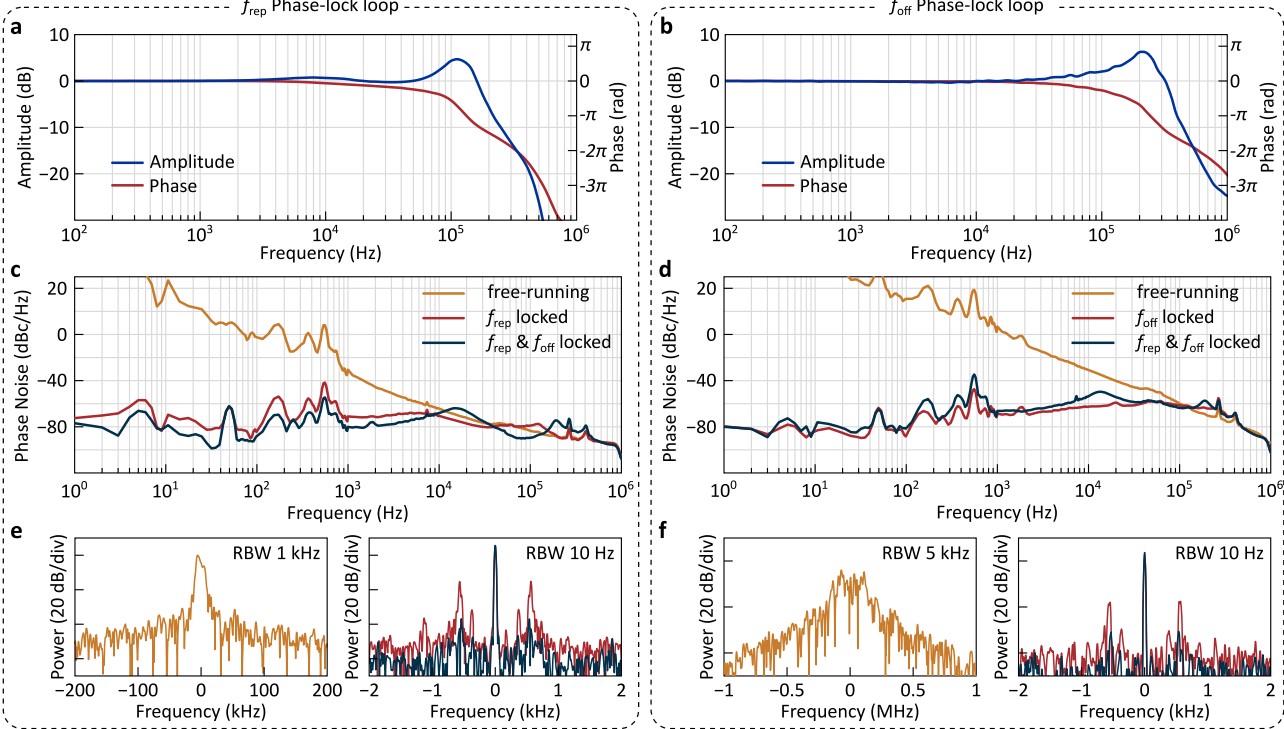

**Fig. 4 | Characterisation of the phase-locked loops. a, b** Closed-loop frequency response of the repetition rate $f_{rep}$ (**a**) and offset beat note $f_{off}$ (**b**) phase-lock loops. **c, d** Single-sideband phase noise of $f_{rep}$ (**a**) and $f_{off}$ (**b**) in the free-running, partially locked and fully locked states. **e, f** Repetition rate beat notes (**e**) and offset beat notes (**f**) corresponding to the respective state shown in (**c**) and (**d**). Note the difference in scale of the frequency axis between the free-running (left) and locked (right) states.

the Supplementary Information. As the corresponding control matrix is diagonalisable with non-zero diagonal elements, the two actuators enable independent control of both degrees of freedom of the SIL microcomb ($\nu_p$ and $f_{rep}$). As we show in the SI, Section 3, the actuators are linear over a large actuation range (exceeding what is needed for phase-locking by orders of magnitude) and hence enable robust operation even under changing environmental conditions.

Depending on the application scenario, a frequency comb may be stabilised to different references, such as two lasers for frequency division and clock operation[35,36], or a repetition rate and self-referencing signal[22,37–41] to provide a phase-coherent radio-frequency-to-optical link.

Figure 2a shows the experimental setup for proof-of-concept stabilisation and characterisation of the microcomb. Specifically, we validate the capability of our system to achieve full-phase stabilisation by comparing our microcomb against a conventional optical frequency comb (OFC). The 1 GHz repetition rate of the conventional OFC is phase-locked to a 10 MHz signal from a GPS disciplined Rb-clock[42]. As Fig. 2b illustrates, an error signal for stabilisation of $\nu_p$ is generated by recording the offset beatnote $f_{off}$ between the central microcomb line $\nu_p$ and the closest line of the reference OFC[21,37] (note that this offset is not to be confused with the carrier-envelope offset frequency). To obtain a repetition rate error signal, we utilise electro-optic phase-modulation (modulation frequency $f_m \approx 17.5$ GHz) of the central comb line and detect the beating $f_b = f_{rep} - 17 \times f_m \approx 200$ MHz between 17th modulation sideband and the first sideband of the microcomb[43]. Both beat notes are then frequency-divided down to approximately 10 MHz, and the error signals are extracted through phase detection with respect to the 10 MHz Rb-clock signal (all microwave sources and recording devices are also referenced to the 10 MHz signal from the Rb-clock). The phase-locked loops (PLLs) are implemented using two conventional off-the-shelf proportional-integral-derivative (PID) controllers, acting onto the laser diode's driving current $I_p$ and the

microheater current $I_h$ for the offset $f_{off}$ and repetition rate $f_{rep}$ stabilisation, respectively. As follows from Table 1, alternative configurations of the PLLs are possible, e.g. swapping the actuators or simultaneously using both actuators for both degrees of freedom to diagonalise the control matrix, which would, however, involve specifically designed PID controllers (e.g., via a field-programmable gate array, FPGA).

Finally, an independent out-of-loop validation of the microcomb's phase-stability is performed by recording the beat note $f_{ol} = 2 \times f_{rep} + f_{off} - 601 \times 1$ GHz between the second sideband of the microcomb and the 601th sideband of the reference OFC. Impacted by both phase locks, the out-of-loop measurement is a key metric in evaluating the overall system performance.

## Experiments

The successive initiation of both PLLs is shown in Fig. 3a where the spectrogram of the out-of-loop beat note is presented. While activating the offset lock already substantially enhances the stability of the out-of-loop beat note (at -1.25 s in Fig. 3a), the remaining fluctuations are only suppressed with the additional activation of the repetition rate lock (at -2.8 s in Fig. 3a). Thus, the two high-bandwidth actuators and the extended detuning range, reliably obtained through synthetic reflection, enable phase stabilisation of the microcomb.

When the microcomb is phase stabilised, the phase excursion in the signals $f_{rep}$, and $f_{off}$ are restricted to a limited interval by the PLLs. This restriction also implies that the phase excursion in the signal $f_{ol}$ are bounded, as long as the reference OFC is phase-stabilised and differential variations of the in-loop and out-of-loop detection paths are negligible.

To get an insight into the nature of the phase excursions, we record the frequency evolution of the out-of-loop beat note $f_{ol}$ with a gate time $\tau = 1$ s and without dead time between the non-overlapping samples (the frequency is extracted from the signal's quadratures, see

Methods). The measured frequencies (shifted to zero-mean) are displayed in Fig. 3b and the corresponding histogram is presented in Fig. 3c (standard deviation of 4.25 Hz). The scatter of the frequency values (and hence the phase deviations) is well approximated by a Gaussian distribution, indicating random noise processes as their origin, as expected for a phase-locked state. Similar data can be obtained for $f_{rep}$ and $f_{off}$ but are not shown here.

Robust and tight phase-locking also manifests itself in the overlapping Allan deviations (OADs) of $f_{rep}$, $f_{off}$ and $f_{ol}$, which we show in Fig. 3d (see Methods). For sufficiently long averaging gate time $\tau$, the OAD is expected to scale proportional to $\tau^{-1}$ as phase excursions are bounded by the PLLs. Indeed, the observed scalings of the OADs— $\tau^{-0.997}$, $\tau^{-0.996}$ and $\tau^{-0.922}$ for $f_{rep}$, $f_{off}$ and $f_{ol}$ respectively—follow the expectation for a phase lock. Importantly, the scalings of the OADs are clearly distinct from the $\tau^{-0.5}$ scaling characteristic of an unbounded random walk of the phase, which would result from random cycle slips in an imperfect phase lock. As such, the OADs of $f_{rep}$ and $f_{off}$ demonstrate the successful implementation of the phase locks, and the OAD of $f_{ol}$ provides an independent out-of-loop validation. For comparison, we also show the OAD of the free-running $f_{rep}$ signal, which, due to uncontrolled frequency drifts, results in an increasing OAD.

Complementing the Allan deviation measurement, we show in Fig. 3e the phase noise of the out-of-loop beatnote $f_{ol}$ to illustrate the impact of the phase lock. Consistent with Fig. 3a, activating the $f_{off}$ lock leads to a first reduction of the phase noise, and activation of both locks lowers the phase noise even more; the phase noise is limited at low frequencies by the phase noise of the reference OFC[42]. The crossing point of the free-running and fully-locked phase noise traces reveals a remarkably high locking bandwidth of larger than 100 kHz that is implemented via the diode current and the simple microheater.

To provide more insights into the locking actuators, we record the closed-loop frequency responses of the repetition rate and offset PLLs as shown in Fig. 4a and b (with the respective other degree of freedom unlocked). Bandwidths of over 100 kHz and 300 kHz, respectively, are achieved for the microheater-based repetition rate actuator and the laser diode-based offset actuator. Both actuators allow for broadband noise suppression, as can be observed from the phase noise of the repetition rate $f_{rep}$ and offset $f_{off}$ signals (Fig. 4c, d) and their corresponding beat notes (Fig. 4e, f). We compute the residual phase modulation (PM) on $f_{rep}$ and $f_{off}$ in the fully-locked state (obtained by integrating the phase noise down to 1 Hz), yielding a root mean square (RMS) residual PM of 0.13 rad and 0.86 rad respectively. Considering the 20x prescaler in the PLLs, the residual PM values are at or below the milliradian level, confirming the tight phase lock. Despite the cross-talk between both actuators, no substantial degradation of the locking performance is observed when both degrees of freedom are locked simultaneously. In the Supplementary Information (SI), we present additional details on the actuator dynamics, linearity, and frequency response.

## Discussion

In conclusion, we demonstrate full phase-stabilisation of a self-injection-locked microresonator frequency comb, and validate its performance through comparison with a conventional mode-locked laser-based frequency comb. Based on a photonic-chip integrated microresonator, our system operates solely on CMOS-compatible driving and control voltages. The control actuators—comprising a laser diode and microheater—achieve a feedback bandwidth exceeding 100 kHz. In conjunction with synthetic reflection, our approach enables robust phase-locking of the microcomb to external frequency references in an unprecedentedly compact form factor (see SI for a comparison).

Our microcomb source (Fig. 1b) is implemented within a sub-1 mm² footprint and does not require the use of tabletop lasers, amplifiers, or high-voltage actuators. Future work could potentially leverage on-chip pulse amplification[44] and integrated f-2f interferometry[45–47] to achieve chip-scale self-referencing[22,37–41], thereby implementing a phase-stable radio-frequency to optical link. Stabilising our system to two atomic clock transitions would result in a compact optical clock, and the capability to achieve phase coherence between multiple sources can be instrumental for synchronisation of large scale facilities or communication networks.

As such, our demonstration establishes a novel, small-footprint, low-complexity, low-cost, and CMOS-compatible frequency comb source for demanding metrological applications, including those in portable, mobile, and integrated settings. The presented results may also inform the design of other chip-integrated light sources, such as rapidly tunable lasers or optical parametric oscillators.

## Methods

### Sample fabrication

The samples were fabricated commercially by LIGENTEC SA using ultraviolet stepper optical lithography. The microresonator ring radius of 75 μm corresponds to an FSR of 300 GHz, while a waveguide width of 1600 nm and a waveguide height of 800 nm provide anomalous group-velocity dispersion (difference between neighbouring FSRs at the pump frequency, $D_2/2\pi \approx 9$ MHz). A coupling gap of 500 nm between the ring and bus waveguide ensures the resonator is critically coupled. Synthetic feedback to the driving DFB diode laser is provided by a nano-patterned corrugation, the amplitude and period of which were chosen to achieve a forward-backwards coupling rate $\gamma/2\pi \approx 145$ MHz at the pump wavelength of ~1557 nm[20]. All modes, including the pump mode, exhibit a high quality factor of $Q \approx 1.5 \times 10^6$.

### Frequency stability measurements

To measure the long-term stability of the microwave signals, we record the beat note's in-phase and quadrature (I/Q) components using the built-in I/Q-analyser of an electronic spectrum analyser (Rohde & Schwarz FSW26). The phase is then extracted from the I/Q data, from which the overlapping Allan deviation is computed using the *Allan-Tools* python module implementing the NIST standards[48]. Frequency counts are obtained by evaluating the finite differences of the extracted phase over the gate time. Spectrograms, spectra, and phase noises are calculated similarly from IQ data.

### Frequency response measurement

To record the closed-loop frequency response of each of the actuators (Fig. 4a, b), a modulation tone is added to the error signal at the input of the PID controller while locked, effectively modulating the set-point. The amplitude and phase of the error signal are then recorded as a function of the modulation frequency using a vector network analyser.

## Data availability

The processed data is available on Zenodo: https://doi.org/10.5281/zenodo.12686076.

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

## Acknowledgements

This project has received funding from the European Research Council (ERC) under the EU's Horizon 2020 research and innovation program (grant agreement No 853564), from the EU's Horizon 2020 research and innovation program (grant agreement No 965124) and through the Helmholtz Young Investigators Group VH-NG-1404; the work was supported through the Maxwell computational resources operated at DESY.

## Author contributions

T.W., A.U., and T.H. conceived the experiment. T.W. and A.U. designed the setup and the photonic chip, performed the experiments, and analysed the data. T.V. developed and operated the reference comb. B.R. supported the design of the resonator and the experiments. T.H. supervised the work. T.W., A.U., and T.H. prepared the manuscript with input from all authors.

## Funding

## Competing interests

The authors declare no competing interests.

## Additional information

**Peer review information** : *Nature Communications* thanks the anonymous, reviewer(s) for their contribution to the peer review of this work. A peer review file is available.

