## [Peer Review File · Nature Communications]

REVIEWER COMMENTS

Reviewer #1 (Remarks to the Author):

The work of the authors is dedicated to the problem of stabilizing the source of a frequency comb generated by a Kerr nonlinearity microresonator, which is pumped by a semiconductor laser in the mode of self-injection locking. Optical frequency combs have become an extremely convenient tool for fundamental metrology and practical applications such as lidars and high-resolution spectrometers. The use of self-injection locking makes such a source relatively simple, and integrated technology allows for its compactness. However, frequency drift of the microresonator and laser parameters can lead to instability of the central frequency and the FSR, which is unacceptable for demanding applications.

In this work, the authors used a corrugated resonator previously proposed and implemented by them to achieve an elegant method of simultaneous stabilization of these two parameters. Judging by the presented graphs, the absolute frequency variations of the comb lines were reduced by more than an order of magnitude over timescales of about a second. The reduction in phase noise was more than 50 dB at frequencies up to several kHz, which is an impressive result.

My remarks on the work:

1. Figure 3a shows the Allan variance for the beatnote signal with an external reference source in the operational stabilization mode. It would be more informative to also present a similar dependence on this graph with the stabilization turned off.
2. In Figure 3a, the Allan variance is only shown for averaging times less than 100 seconds, and there is no characteristic rise at longer times. Since the proposed method should be effective at longer times, it would be desirable to see the results for a longer time at which the dependence changes.
3. The paper claims that the level of phase noise at frequencies below 1 kHz is determined by the digital PLL system (a reference to the authors' earlier work [41] is given), which is obviously not a fundamental limit. I would suggest the authors analyze the achievable phase noise limits in this system.
4. External frequency combs locked to an atomic transition frequency were used for stabilization in this work. This is a complex method that requires a lot of auxiliary equipment. It allowed the authors to demonstrate the effectiveness of their proposed method but is obviously not suitable for practical use, as it undermines its advantages such as compactness and simplicity. The authors should add a discussion of possible practical implementations of this method, including ways to fully integrate it on a single platform.

Taking these comments into account, I recommend this work for publication.

Reviewer #2 (Remarks to the Author):

The paper "Phase-stabilised self-injection-locked microcomb" by Wildi et al. presents the demonstration of full phase stabilization of a self-injection-locked microcomb, achieved through the implementation a synthetic reflection within the microresonator. The authors use current tuning of the laser and the microresonators integrated heater to control the comb's offset and line spacing frequencies, achieving a locking bandwidth of ~ 100 kHz for both actuators.

The paper exhibits good technical quality, with a well-presented methodology and clear experimental results. However, it is overly technical for the selected journal, lacking major novelty compared to existing literature. The utilization of synthetic reflection [ref. 20 by the same authors] and heater-controlled microresonators [ref. 26] has been previously published. The stabilization scheme, while combined with self-injection locking, does not significantly differentiate itself from existing methods as two independent PLLs are eventually used. Moreover, the paper lacks an explanation of the physical mechanism behind the tuning of the comb's degrees of freedom in the SIL regime. Therefore, I do not recommend publication in Nature Communications but suggest submission to a more specialized journal.

Specific comments and questions:

Is the tuning of f_{rep} and f_{off} uniform with the change in currents? The tuning slopes can vary with the detuning [<https://doi.org/10.1038/ncomms14869>, <https://doi.org/10.1103/PhysRevA.95.043822>, <https://doi.org/10.1103/PhysRevLett.121.063902>] due, for example, to asymmetries in the spectrum and the amount of energy in the resonator. Is the f_{rep} rate variation arising mainly from the Raman effect (i.e., detuning change effect) or from the thermal variation in the ring (physical variation of the FSR)? How is the crosstalk between f_{rep} and f_{off} managed? Does it cause instability? Is the implemented scheme working because of the relatively small crosstalk? The authors mention that the system's actuation could be diagonalized, but do not elaborate on how this could be achieved in practice. I suppose the FPGA feedback control could be leveraged to its full potential here.

Does the crosstalk between the actuators impact the general stability of the lock? For how long was the stabilized system running?

Laser diodes such as DFBs are known to feature a typical modulation bandwidth in the range of GHz. What limits the bandwidth to ~ 300 kHz in the present case? Does the SIL effect drastically reduce the tuning speed of the laser?

Please, indicate the measurement limits of the phase noise measurement system when showing phase noise data. Is the phase noise of the out-of-loop signal limited by the multiplied noise of the reference oscillator? Does the digital PLL introduce a significant amount of noise?

Please explain how the frequency responses were measured in the closed-loop system, or provide a reference. Do you inject a modulation tone while the system is locked? Where is the response signal measured? Does the locking modify the response?

Reviewer #3 (Remarks to the Author):

This paper discusses the full phase-stabilization of a self-injection-locked microresonator frequency

comb, leveraging a silicon nitride photonic crystal ring resonator (PhCR) and integrated microheater for control. While the experiments are well-designed and the figures effectively illustrate the findings, the paper lacks detailed exploration of new physics. Specifically, it misses crucial details on the PhCR's design and thermal stability, lacks in-depth discussion on the tuning process of actuators for phase stabilization, does not address the system's environmental sensitivity, omits coupling loss information between the DFB laser and the butt coupler, and lacks insights on long-term phase stability. Additionally, a comparative analysis with previous microcombs in terms of performance, efficiency, and cost is missing, questioning the novelty of this work. Given these gaps, particularly in the context of contributing new physics or significant novelty, I do not recommend publication in Nature Communications without substantial revisions to address these issues.

The Reviewers' comments are in black.

The authors' replies are in blue.

Descriptions of revisions are underlined.

We thank all three Reviewers for their careful assessment of our work, insightful comments, and constructive criticism. The Reviewers' comments have led to a substantially revised manuscript with improved clarity and additional data. A new **Supplementary Information (SI)** provides supporting material and insights. We hope the revisions, along with our detailed reply below, satisfactorily address all concerns raised by the Reviewers.

In the manuscript, all revisions and additions are highlighted in blue.

Reviewer #1 (Remarks to the Author):

The work of the authors is dedicated to the problem of stabilizing the source of a frequency comb generated by a Kerr nonlinearity microresonator, which is pumped by a semiconductor laser in the mode of self-injection locking. Optical frequency combs have become an extremely convenient tool for fundamental metrology and practical applications such as lidars and high-resolution spectrometers. The use of self-injection locking makes such a source relatively simple, and integrated technology allows for its compactness. However, frequency drift of the microresonator and laser parameters can lead to instability of the central frequency and the FSR, which is unacceptable for demanding applications.

In this work, the authors used a corrugated resonator previously proposed and implemented by them to achieve an elegant method of simultaneous stabilization of these two parameters. Judging by the presented graphs, the absolute frequency variations of the comb lines were reduced by more than an order of magnitude over timescales of about a second. The reduction in phase noise was more than 50 dB at frequencies up to several kHz, which is an impressive result.

We thank the Reviewer for the careful assessment of our work, the constructive feedback and the recommendation for publication (see end of report). Below, we reply to all remarks made by the Reviewer and detail the modifications and additions to the manuscript we have made in response.

My remarks on the work:

1. Figure 3a shows the Allan variance for the beatnote signal with an external reference source in the operational stabilization mode. It would be more informative to also present a similar dependence on this graph with the stabilization turned off.

We thank the Reviewer for the suggestion and agree that contrasting Allan deviations for the stabilised and free-running (stabilisation turned off) state would be informative. To illustrate the difference between the free-running and the stabilised state, we have thus added a new trace to the figure, showing the Allan deviation of f_{rep} in the free-running regime. As opposed to the stabilised state, where the Allan deviation averages down with longer gate times, the free-running state exhibits a markedly larger Allan deviation that increases with gate time (At the longest measured gate time, the difference between the Allan deviation in the stabilised and free-running state reaches 9 orders of magnitude).

Note that we do not show the Allan deviations for the free-running f_{ol} and f_{off} signals as the respective frequency excursions are even larger than that of f_{rep} . Besides not adding additional value, f_{ol} and f_{off} , in practice, cannot be continuously recorded as it would require sampling the I/Q signals with > 100 MHz sampling rate over prohibitively long temporal intervals.

2. In Figure 3a, the Allan variance is only shown for averaging times less than 100 seconds, and there is no characteristic rise at longer times. Since the proposed method should be effective at longer times, it would be desirable to see the results for a longer time at which the dependence changes.

We thank the Reviewer for this important remark, which we would like to discuss in more detail:

Generally, the Allan deviation quantifies the frequency stability on long time scales. For instance, it is often used to compare the relative stability of two independent (free-running) oscillators. Even very stable oscillators will start diverging after some time. The time scale, on which this di-

vergence becomes relevant, is the gate time for which the Allan deviation shows a characteristic rise.

In our case, we use the Allan deviation not for the comparison of two independent free-running oscillators but for phase-locked oscillators, i.e. where the feedback loop actively synchronises one oscillator to the other. In this scenario, assuming a working phase lock, the Allan deviation of the *in-loop* beat note is expected to average down indefinitely with a scaling close to τ^{-1} : the maximum phase excursion is bounded by the phase lock, and as the averaging time (gate time) increases, the corresponding frequency deviation decreases. For an imperfect phase lock with cycle slips, the phase would undergo a random walk, resulting in a $\tau^{-1/2}$ scaling of the Allan deviation (which we do not observe). Hence, the fact that the in-loop beatnote averages down close to τ^{-1} validates the performance of the phase-lock.

The *out-of-loop* beatnote and its Allan deviation can provide an independent validation of the phase-lock. Indeed, its scaling close to τ^{-1} confirms the phase lock. However, we note the τ^{-1} scaling in the out-of-loop beatnote can only be observed if uncontrolled differential phase drifts between the in-loop and out-of-loop detection paths are negligible. As the Reviewer suggest, for very long gate times the out-of-loop Allan deviation can be expected to rise due to differential drifts between the in-loop and out-of-loop detection paths due changing laboratory conditions, material ageing, etc. Our measurements indicate that over the time scales considered here, such differential drifts are negligible and that the entire setup, including the differential detection path, is phase-stable. Thus, the out-of-loop signal can here serve as an independent validation of the phase lock. To clarify the interpretation of the Allan deviation, we have completely revised the corresponding section in the manuscript, in particular with regards to interpreting the slope of the Allan deviation and the meaning of the in-loop and out-of-loop Allan deviations.

In our manuscript, we initially used a gate time of 100 s, following the standard set by previous works on microcomb stabilisation [1, 2]. Similar to the present case, those previous works did not observe a rise in the Allan deviation. Nevertheless, we welcome the request for longer duration data and have extended our gate time by a factor of 3, now up to 300 s (c.f. revised manuscript, Figure 3c). We still do not observe any deviation from the τ^{-1} scaling. We suspect this is due to a (desirable) high level of commonality between the in-loop and out-of-loop paths; for instance, impacts of slowly changing laboratory temperature cancel out, and long-term factors such as component ageing do not yet factor in.

1. Del’Haye, P., Arcizet, O., Schliesser, A., Holzwarth, R. & Kippenberg, T. J. Full Stabilization of a Microresonator-Based Optical Frequency Comb. *Physical Review Letters* **101**, 053903 (2008).
 2. Brasch, V., Lucas, E., Jost, J. D., Geiselmann, M. & Kippenberg, T. J. Self-Referenced Photonic Chip Soliton Kerr Frequency Comb. *Light: Science & Applications* **6**, e16202–e16202 (2017).
3. The paper claims that the level of phase noise at frequencies below 1 kHz is determined by the digital PLL system (a reference to the authors’ earlier work [41] is given), which is obviously not a fundamental limit. I would suggest the authors analyze the achievable phase noise limits in this system.

Indeed, as the Reviewer states, the phase noise measured below 1 kHz is limited by our reference optical frequency comb (OFC) and the digital PLL used for its stabilisation; see reply to question 8 of Reviewer #2 for technical details. As such, the measured phase noise levels in Figure 3e of the manuscript provide an upper bound to the phase noise of the microresonator comb lines. This is shown in Figure 1 (in this document) by the new grey trace showing the repetition rate phase noise of the OFC (phase noise multiplied to account for the respective frequency interval). Indeed, the measured phase noise follows the expected limitation imposed by the reference OFC.

A reasonable estimate of the achievable phase noise level of the stabilised microcomb can be compiled from the in-loop offset and repetition rate signal phase noises (in the fully locked state). Assuming a noise-free reference comb (i.e. not limited by the measurement apparatus), the noise on f_{rep} and f_{off} fully define that of the out-of-loop beat note. In Figure 1 (in this document), we have added this estimated ‘true’ phase noise so that it can be compared to the measured phase noise of the out-of-loop signal in the fully-locked state in Figure 1 (light blue trace). This noise estimate follows the measured data for frequencies >1 kHz, while below 1 kHz, the reference OFC is clearly the limiting factor. From this, we conclude that the estimated achievable phase noise

Figure 1: Single-sideband phase noise of the out-of-loop beat note f_{ol} .

is $\lesssim 60$ dBc/Hz below 1 kHz offset. To make the current measurement limitation more visible to the reader, we have added the multiplied repetition rate noise of the reference mode-locked optical frequency comb to the revised manuscript (grey trace, Figure 3e). We have, however, not included the estimation based on the in-loop data in the manuscript, as we do not want to (falsely) suggest to the reader (who may only glance at the figure) that we had indeed measured such phase noise.

For clarity, we emphasise that the focus of our work is not a demonstration of low-phase noise (it is only a by-product) but rather the phase-locking, which ensures long-term phase stability between the microcomb and those external references we use for locking. The intention of showing the phase noise in this present case is to provide additional information on the phase locks: while the Allan deviation shows long-term phase stability, the phase noise data demonstrates that the locking feedback loops work indeed up to a bandwidth of 10^5 Hz (where the free-running and fully locked curves intersect). To clarify the meaning of the phase noise graph, we have revised the corresponding section in the manuscript and explained that it provides a measure of the effective locking bandwidth, which is remarkably high for a heater-based actuator.

4. External frequency combs locked to an atomic transition frequency were used for stabilization in this work. This is a complex method that requires a lot of auxiliary equipment. It allowed the authors to demonstrate the effectiveness of their proposed method but is obviously not suitable for practical use, as it undermines its advantages such as compactness and simplicity. The authors should add a discussion of possible practical implementations of this method, including ways to fully integrate it on a single platform.

We thank the Reviewer for this remark and suggestion for an additional discussion. Before we provide this discussion, we establish a common starting point, although we believe these clarifications are likely unnecessary, given the Reviewer’s background.

One way to convincingly show the metrology-grade performance of a chip-scale comb is to validate it against a conventional and established table-top mode-locked laser frequency comb. By effectively locking together one comb line of each comb and synchronising both combs’ line-spacings to a common 10 MHz signal, we achieve full phase-coherent synchronisation between both combs (In our implementation, the common 10 MHz signal is provided by a Rubidium clock; this is not critical, and another 10 MHz source may be used). This synchronisation is then validated by means of an independent out-of-loop heterodyne beatnote between both combs. Therefore, to validate phase-locking and metrology-grade performance, we indeed require a complex setup involving, in particular, a conventional frequency comb. However, as also the Reviewer points out, this setup’s purpose is to validate the system’s properties; it is neither a prerequisite nor a representation of how the system would be used in an application scenario.

To illustrate this, we provide examples of applications (that do not require the conventional frequency comb): One possible application scenario of this present microcomb could be the implementation of an optical clock by locking two of its lines to the rubidium clock transition at 384.23 THz and 377.11 THz, which are both (via two-photon absorption or frequency doubling) within the comb’s bandwidth. In another application scenario, our comb could be synchronised to another integrated comb source, which can be useful for time distributions in large-scale facilities

(accelerators, radio telescopes), as well as ground and space-based communication networks. Finally, with on-chip pulse amplifiers [1] and highly efficient chip-based octave spanning supercontinua and offset detection, fully integrated f-2f interferometers [2–4] are within reach. In conjunction with our demonstration of the first chip-scale microcomb system that is fully phase-stabilised, this would enable truly chip-scale radio-frequency-to-optical links.

We thank the Reviewer for pointing out the missing discussion. To address this point, we have clarified in the conclusion that the conventional optical frequency comb is only for validation purposes. We also added application scenarios that do not require an external conventional comb, along the lines above, to the conclusion.

1. Gaafar, M. A. *et al.* *Femtosecond Pulse Amplification on a Chip* 2023. arXiv: 2311.04758 [physics].
2. Carlson, D. R. *et al.* Self-referenced frequency combs using high-efficiency silicon-nitride waveguides. *Opt. Lett., OL* **42**. Publisher: Optica Publishing Group, 2314–2317 (2017).
3. Okawachi, Y. *et al.* Chip-based self-referencing using integrated lithium niobate waveguides. *Optica, OPTICA* **7**. Publisher: Optica Publishing Group, 702–707 (2020).
4. Obrzud, E. *et al.* Stable and Compact RF-to-optical Link Using Lithium Niobate on Insulator Waveguides. *APL Photonics* **6**, 121303 (2021).

Taking these comments into account, I recommend this work for publication.

We thank the Reviewer for recommending publication, once the Reviewer’s comments have been taken into account. We greatly appreciate the detailed and helpful comments, which have allowed us to significantly increase the clarity and appeal of our manuscript. We hope that our revision and explanations satisfactorily address all comments raised by the Reviewer.

Reviewer #2 (Remarks to the Author):

The paper "Phase-stabilised self-injection-locked microcomb" by Wildi et al. presents the demonstration of full phase stabilization of a self-injection-locked microcomb, achieved through the implementation a synthetic reflection within the microresonator. The authors use current tuning of the laser and the microresonators integrated heater to control the comb's offset and line spacing frequencies, achieving a locking bandwidth of 100 kHz for both actuators.

The paper exhibits good technical quality, with a well-presented methodology and clear experimental results. However, it is overly technical for the selected journal, lacking major novelty compared to existing literature. The utilization of synthetic reflection [ref. 20 by the same authors] and heater-controlled microresonators [ref. 26] has been previously published. The stabilization scheme, while combined with self-injection locking, does not significantly differentiate itself from existing methods as two independent PLLs are eventually used. Moreover, the paper lacks an explanation of the physical mechanism behind the tuning of the comb's degrees of freedom in the SIL regime. Therefore, I do not recommend publication in Nature Communications but suggest submission to a more specialized journal.

We thank the Reviewer for the careful evaluation of our work, recognising the quality, methodology, and clarity of the results. We welcome the valuable comments that we will address in our reply. Further, we understand that (1.) based on previous work (refs. 20 and 26) and (2.) the use of two PLLs (like in previous systems), the Reviewer challenges the novelty of our work. In addition, (3.) the Reviewer would like to see a better explanation of the physical processes involved in tuning the comb's degrees of freedom. We appreciate this criticism and have substantially revised the manuscript to address all comments, as we will detail below. We will also explain in more detail why we believe our work is novel, distinct from previous work, and indeed of interest to the wide audience of Nature Communications.

1. Differentiation with respect to ref. 20 and ref. 26

Ref. 20 reports on synthetic reflection as a novel method to reliably and deterministically operate a self-injection locked microresonator comb in the low-noise single soliton pulse regime without relying on random sample properties. While we leverage the insights of ref. 20 and build on top of it, we emphasise that ref. 20 does not discuss or even mention phase stabilisation. As such, ref. 20 and the present work are fundamentally distinct.

Ref. 26 reports on the thermal tuning of a microresonator by means of a microheater to achieve a soliton state. This is a milestone accomplishment. However, we would like to stress that achieving a phase-locked loop via a thermal heater is significantly more challenging than the approximate thermal ramp required to initiate soliton states. Ref. 26 neither investigates the actuation speed of the heater nor its phase response, both critical parameters for establishing a control loop. Prior to our work, it was completely unclear whether a microheater, despite its simplicity, could enable phase-locking.

2. Differentiation with regard to previous work that also uses two PLLs.

Metrology-grade comb sources generally consist of a pulse source and, critically, the ability to control the repetition rate and offset frequency of the pulse train well enough to establish a phase lock between the two comb lines and two external frequency references. Such fully phase-locked systems represent the backbone of modern time and frequency metrology (optical clocks, optical precision spectroscopy, time transfer and synchronisation, pump-probe experiments, etc.); however, they were previously only attainable in table-top frequency comb lasers or microresonators in conjunction with table-top scientific laser systems (such as low-noise external cavity diode lasers, erbium-doped fibre amplifier, acousto-optic modulators or single-sideband generators). Our work changes this and provides the first chip-scale comb source that can be phase-locked to external references.

With regard to using two phase-locked loops (PLLs), we note that phase stabilisation of a frequency comb's two degrees of freedom generally requires two PLLs. Therefore, the critical question here is not whether two PLLs are used but whether the system offers two suitable linearly independent and fast actuators that permit stabilisation of the comb. Our work demonstrates this with actuators that are, for the first time, implemented on the chip scale.

Below, we would like to provide more detail to better put our demonstration into context:

Early work demonstrated that full phase control in microcombs could be achieved by pumping the system with a scientific low-noise external cavity diode laser (ECDL) in combination with an erbium-doped fibre amplifier (EDFA). The ECDL defined the central comb line and enabled its actuation via combined piezo and current modulation. Modulation of the EDFA’s pump current enabled rapid power actuation, which would modulate heating by absorption in the resonator, impacting the repetition rate primarily via the thermo-refractive effect. While effective, this approach is neither compact nor scalable, and the needed equipment has confined microcombs to research laboratories and niche applications.

Excitingly, recent years have shown a disruptive development in microresonator combs towards self-injection locked systems. In these self-injection locked systems, a low-cost semiconductor laser chip replaces the table-top ECDLs and EDFAs, resulting in a compact system. Moreover, self-injection locking drastically reduces the operational complexity of microcombs, as the self-injection locking mechanism regulates the laser tuning and guarantees deterministic operation. Such self-injection locked systems have opened an attractive and credible route for microcombs to venture into their envisioned cross-disciplinary applications. However, moving towards a self-injection-locked (SIL) system comes at a cost. The central frequency is now controlled by the self-injection locking mechanism, where the laser and resonator form an inseparable system. Established methods for phase stabilisation are no longer applicable. With the diode laser pump current being the only actuator, only one of the two degrees of freedom can be stabilised. So, while SIL enabled compactness and scalability, the ability to phase-stabilise the system, a key property of frequency combs, is lost.

In our work, we demonstrate the essential but previously lacking ability of full-phase control in such compact chip-scale systems utilising only a simple microheater. Implementing a metrology-grade system with such simple means is all but obvious – and of high relevance to applications.

While the diode laser current provides a fast actuator (bandwidth $\gg 10$ MHz), the microheater’s 3 dB modulation bandwidth is only ~ 5 kHz. As both actuators are coupled to both degrees of freedom, the system’s bandwidth is limited by the slowest actuator (the heater). Our work shows that the PLL driving the microheater can indeed be arranged to reach an effective actuation bandwidth of > 100 kHz; this has not been shown before. Still, this locking bandwidth would be largely insufficient to phase-lock a DFB diode laser. This means that implementing a conventional approach to phase stabilisation via the diode laser and the microheater would likely not work. In the case of self-injection locking, the situation changes as the self-injection results in a drastic collapse of the linewidth of the driving laser. Through this effect, the bandwidth requirements for the actuators are significantly reduced, and phase-locking becomes feasible.

Microheaters are standard components in micro-photonic foundries, and our approach can hence be applied to any resonator platform, without the need for high-voltage piezo actuation or electro-optic materials. The simplicity of our approach and its cross-compatibility with any resonator platform establishes microcombs not only as low-cost and compact sources but, indeed, as systems that can meet the demanding requirements of phase-coherent precision metrology. In our view, this is not incremental and represents a major milestone for optical precision metrology.

To better place our work into context and highlight its novelty, we have revised a large part of the introduction as well as the conclusion. Moreover, we have added a table to the SI that compares our work to previous demonstrations of phase-stabilised microcombs. This table shows that our work shrinks phase-stabilised combs by more than three orders of magnitude and reduces their estimated cost by more than two orders of magnitude.

Further, in the SI, we have added data on the open-loop frequency response of the microheater, showing its ~ 5 kHz 3 dB-bandwidth, illustrating that achieving more than 100 kHz effective locking bandwidth is all but obvious. We now also explicitly state in the manuscript the effect of laser linewidth narrowing in SIL, which critically relaxes the actuation bandwidth requirement for phase-locking.

3. Improved explanation of the physical mechanism.

We welcome the request for more information on the underlying physical mechanism at work. To address this request, we provide now in Section 1 & 2 of the SI a detailed analysis of the underlying

physical mechanism for the offset tuning (impacted by diode current and heater current) and for the repetition rate tuning (impacted by diode current and heater current), which is mediated by stimulated Raman scattering (SRS). Moreover, in Section 3 of the SI, we have added new data, including a study on the linearity of both actuators (heater and diode), as well as frequency response characteristics of the microheater. Additional and detailed discussions are presented below in direct response to the specific comments and questions raised by the Reviewer.

As an intermediate conclusion, we demonstrate for the first time full phase-stabilisation in a chip-scale, electrically driven comb system. The ability to achieve full phase-stabilisation is a key property of a frequency comb and crucial for comb-based precision metrology (spectroscopy, timing, etc.). Our result marks the first implementation of a microcomb with these key characteristics of a metrological comb in a chip-scale setting. This is a major qualitative and non-incremental achievement for optical metrology and chip-scale frequency combs. We believe our results will spark exciting developments and have a sustained impact not only on the microresonator community but indeed on those who may utilise microcombs in cross-disciplinary applications.

Specific comments and questions¹:

1. Is the tuning of f_{rep} and f_{off} uniform with the change in currents? The tuning slopes can vary with the detuning [<https://doi.org/10.1038/ncomms14869>, <https://doi.org/10.1103/PhysRevA.95.043822>, <https://doi.org/10.1103/PhysRevLett.121.063902>] due, for example, to asymmetries in the spectrum and the amount of energy in the resonator.

We agree with the Reviewer that, generally speaking, the dependence of f_{rep} and f_{off} on both the pump current I_p and heater current I_h is nonlinear. For example, the SIL tuning curve (c.f. Supplemental Information, Figure S1), which describes the relation between laser emission frequency in the absence of self-injection locking (controlled by the diode current) and the laser frequency-to-resonance detuning under self-injection locking, is nonlinear. Also, the power dissipated in the microheater depends quadratically on the heater current.

However, within the operation range of our system, the tuning behaviour can be considered linear: we have recorded the tuning curves $f_{\text{rep}}(I_p)$, $f_{\text{rep}}(I_h)$, $f_{\text{off}}(I_p)$ and $f_{\text{off}}(I_h)$ around the system operation point and added them to the revised manuscript's SI, in Section 3. All tuning curves are monotonic and, to good approximation, linear, which ensures stable locking conditions.

2. Is the rep rate variation arising mainly from the Raman effect (i.e., detuning change effect) or from the thermal variation in the ring (physical variation of the FSR)?

We thank the Reviewer for his question, which sparked us to look into the precise tuning mechanism behind each actuator. As the Reviewer correctly anticipates, both thermal variation of the FSR and the Raman shift (sensitive to detuning) are relevant. We added a detailed description of the physical tuning mechanisms in Section 1 & 2 of the SI. The analytically estimated tuning coefficients agree with those measured experimentally (c.f. manuscript, Table 1).

Specifically, we find that the Raman self-frequency-shift (SFS) is the mechanism behind the diode laser current tuning of the repetition rate (via a change in the pump-to-resonance detuning) with a tuning coefficient of $\partial f_{\text{rep}}/\partial I_p \approx 190 \text{ kHz mA}^{-1}$. On the other hand, while thermal tuning (via the microheater current) also affects the repetition via the Raman SFS, at a rate of $\partial f_{\text{rep}}/\partial I_h \approx 38 \text{ kHz mA}^{-1}$, the main contribution stems from the thermal variation in the microresonators effective optical length at a rate of $\partial f_{\text{rep}}/\partial I_h \approx 325 \text{ kHz mA}^{-1}$ (dominated by thermorefractivity).

3. How is the crosstalk between f_{rep} and f_{off} managed? Does it cause instability? Is the implemented scheme working because of the relatively small crosstalk?

With two input signals and two degrees of freedom, optical frequency comb stabilisation is a multiple-input multiple-output (MIMO) control system problem. As described in the reply to a previous point, our system is effectively linear within the actuation range of our control loops and, therefore, can be diagonalised. Theoretically, system diagonalisation would enable slightly higher performance in suppressing high-frequency noise near the loop's bandwidth, but we expect

¹To structure our reply, we have taken the liberty to group by topic and number the Reviewer's questions.

the improvement to be marginal. In practice, diagonalisation is often unnecessary, as the control loops can efficiently suppress their counterpart’s crosstalk. This allows us to use off-the-shelf PID controllers without the need for custom implementation of the PID controllers with 2-dimensional input and output. We have added comments to the manuscript pointing out that we are using conventional off-the-shelf PID controllers and that diagonalisation would require specifically designed controllers.

The absence of a detrimental effect from the cross-talk between both actuators is further discussed below, where the Reviewer raises a related question.

4. The authors mention that the system’s actuation could be diagonalized, but do not elaborate on how this could be achieved in practice. I suppose the FPGA feedback control could be leveraged to its full potential here.

In practice, as the Reviewer suggests, an FPGA board would be a convenient and flexible way to implement a MIMO controller (it is also compatible with full integration). We now mention in the manuscript FPGAs as a possible means of implementation.

In its simplest form, a MIMO controller could be designed by diagonalising the control matrix, which would remove crosstalk by providing a pair of orthogonal control vectors (i.e. control basis). This approach is straightforward but does not consider the individual frequency response of the two actuators and, therefore, might not be optimal at higher frequencies (>10 kHz).

A more rigorous approach would be to use one of many established system identification methods to first fit a model to our system, such as a state-space model (SSM), an auto-regressive with exogenous input (ARX) model or an auto-regressive moving average with exogenous input (ARMAX) model; and then develop a matching controller. Exploring such feedback schemes, although interesting, is well beyond the scope of this present work.

5. Does the crosstalk between the actuators impact the general stability of the lock? For how long was the stabilized system running?

We thank the Reviewer for inquiring about the effect of the crosstalk and the system’s longevity of the stabilised system: In our system, crosstalk has a negligible effect on lock performance as evidenced in Fig. 4c,d (main manuscript), where the impact on the locked f_{rep} and locked f_{off} phase noise is almost independent of the respective other lock being active or not.

As the actuation range (cf. SI, Section 3) is orders of magnitude larger than what is needed to stabilise the system on short time scales, the stabilised system can run robustly on long time scales. In the manuscript, we now provide this explanation along with a reference to the SI. The current limiting factor is the mechanical drift between the diode chip and microresonator chip, which are, in our setup, not fixed to each other We have added this information to the manuscript. While this enables us to easily study different chip/diode combinations, the absence of product-level packaging leads to gradual misalignment between both components. In practice, this limits the operational lifetime of the soliton comb state to approximately one hour. This is, however, *not* a limitation of the phase-locked loop. It is well known that established packaging methods can mitigate mechanical misalignment if needed [1, 2]; such packaging is, however, not the focus of this present work.

1. Shen, B. *et al.* Integrated Turnkey Soliton Microcombs. *Nature* **582**, 365–369 (2020).
2. Lihachev, G. *et al.* *Frequency agile photonic integrated external cavity laser* 2023. arXiv: 2303.00425[physics].

6. Laser diodes such as DFBs are known to feature a typical modulation bandwidth in the range of GHz. What limits the bandwidth to ~ 300 kHz in the present case?

Indeed, the diode by itself would allow for much faster actuation ($\gg 10$ MHz). However, in our system, both actuators affect both degrees of freedom of the comb. Due to this cross-coupling, the overall bandwidth of our system is restricted to the bandwidth of the slowest actuator, which in this case is the thermal actuator (microheater).

To provide further insight into the thermal actuation of the microcomb, we have added the open-loop frequency response of the microheater in Section 4 of the SI. As can be seen from Fig-

ure S3, we measure for the amplitude response a 3 dB bandwidth of ~ 5 kHz. Such low bandwidth is, at first glance, discouraging and far too low to achieve phase locking. However, the additional measurement of the phase response reveals a π -phase bandwidth extending beyond 100 kHz, indicating that a much higher effective bandwidth should be possible with a suitable feedback loop. In our experiment, we indeed achieved an effective heater actuation bandwidth of more than 100 kHz (c.f. manuscript, Figure 4a). As the heater bandwidth dictates the overall system bandwidth, these insights are critical in achieving both offset and repetition rate locks.

7. Does the SIL effect drastically reduce the tuning speed of the laser?

Yes, when operating in the SIL regime, the frequency tuning per unit change in the drive current is drastically reduced where the exact reduction is dependent on the resonator linewidth, feedback strength, and diode laser power [1]. For the resonator used in our work, the diode laser’s frequency variations are reduced by a factor $\sigma \approx 40$ in the SIL regime, compared to what they would otherwise be if the diode laser were left free-running. This effect is now described in the SI, Section 1. Despite this reduction, the leverage provided by the current tuning is by far sufficient to achieve stable phase-locking. Indeed, the SIL mechanism is crucial as it also narrows the \sim MHz linewidth of the driving diode laser by a factor $\sigma^2 > 1000$, reducing the bandwidth requirement for the actuators and enabling phase-locking.

1. Voloshin, A. S. *et al.* Dynamics of Soliton Self-Injection Locking in Optical Microresonators. *Nature Communications* **12**, 235 (2021).

8. Please, indicate the measurement limits of the phase noise measurement system when showing phase noise data. Is the phase noise of the out-of-loop signal limited by the multiplied noise of the reference oscillator? Does the digital PLL introduce a significant amount of noise?

We thank the Reviewer for the suggestion. We have added the multiplied repetition-rate phase noise of the reference optical frequency comb (OFC) to the revised manuscript. As can be seen in the updated Figure 3e (grey trace), the reference OFC used to obtain the out-of-loop beat note is the limiting factor at frequencies below ~ 1 kHz when the microcomb is fully stabilised.

The underlying technical reason for this limitation is the digital PLL that locks the OFC’s repetition rate to the 10 MHz RF reference: The digital PLL is implemented on an FPGA board with a 12-bit ADC. The smallest phase excursion that can be resolved is roughly 2^{-12} radians. Scaled from 10 MHz to 601 GHz (the frequency interval that underpins our measurement), this represents a phase excursion of ~ 15 rad. Integrating the multiplied phase noise of the reference OFC, we get an RMS residual phase jitter of 15.8 rad, which is in excellent agreement with the previously obtained number, supporting the explanation for the observed limitation.

9. Please explain how the frequency responses were measured in the closed-loop system, or provide a reference. Do you inject a modulation tone while the system is locked? Where is the response signal measured? Does the locking modify the response?

We thank the Reviewer for pointing out the missing description of the method used to measure the actuators’ closed-loop frequency response. The measurement is performed by adding a weak modulation to the error signal at the input of the respective PID controller when locked (i.e. we modulate the system’s setpoint). We then record the amplitude and phase of the error signal (before the addition of the modulation) as a function of the modulation frequency using a vector network analyser. At low frequencies, the PID controllers and actuators easily track the input modulation (0 dB gain), while at higher frequencies, a resonance (*servo bump*) can be observed when a phase delay of $\pi/2$ is reached (c.f. manuscript, Figure 4a and b).

We have added a paragraph in the Methods section of our revised manuscript describing the measurement method for characterising the closed-loop response of the actuators.

In sum, we thank the Reviewer for the valuable comments, which have led to major revision. We believe these revision have strengthened and completed our manuscript. In light of these revision, we hope that the Reviewer can recommend our work for publication.

Reviewer #3 (Remarks to the Author):

This paper discusses the full phase-stabilization of a self-injection-locked microresonator frequency comb, leveraging a silicon nitride photonic crystal ring resonator (PhCR) and integrated microheater for control. While the experiments are well-designed and the figures effectively illustrate the findings, the paper lacks detailed exploration of new physics. Specifically, it misses crucial details on the PhCR’s design and thermal stability, lacks in-depth discussion on the tuning process of actuators for phase stabilization, does not address the system’s environmental sensitivity, omits coupling loss information between the DFB laser and the butt coupler, and lacks insights on long-term phase stability.

We thank the Reviewer for the careful evaluation of our work, recognising the well-designed experiments, and the effective representation of the data in the figures. We also acknowledge that the Reviewer requests substantial revisions before being able to recommend our work for publication (see end of report). Specifically, the Reviewer identifies a lack of details regarding the reported physics, including the design of the resonator, thermal stability, the tuning process of the actuators, environmental sensitivity, coupling loss and long-term phase stability. We welcome all comments and have made substantial revisions to address them, as we detail below:

Fully phase-locked combs represent the backbone of modern time and frequency metrology (optical clocks, optical precision spectroscopy, time transfer and synchronisation, pump-probe experiments, etc.). Previous fully phase-stabilised combs were only attainable in table-top mode-locked lasers or microresonators in conjunction with table-top continuous-wave driving lasers (involving low-noise external cavity diode lasers, erbium-doped fibre amplifier, acousto-optic modulators or single-sideband generators). This meant that any phase stabilised frequency comb would still involve a table-top setup with components costing in excess of 10’000 USD.

Our work changes this and provides the first chip-scale comb source that can be phase-stabilised. The actuators providing phase-locking utilise only low-voltages and are compatible with foundry processes (in contrast to piezo-optic, acousto-optic, or electro-optic actuators). The specific actuators we utilise are the pump diode and the current of a microheater. While the diode laser current provides a fast actuator (bandwidth $\gg 10$ MHz), the microheater’s 3 dB modulation bandwidth is only ~ 5 kHz (This information has been added to the SI, Section 4). Such low bandwidth is, at first glance, discouraging and far too low to achieve phase locking. However, the additional measurement of the phase response reveals a π -phase bandwidth extending beyond 100 kHz, indicating that a much higher effective bandwidth should be possible with a suitable feedback loop (The corresponding data is now presented in the SI, Section 4). Such measurements and characterisation of microheaters for phase-locking are presented in our manuscript for the first time, to the best of our knowledge. Indeed, in our experiment, we achieved a remarkably high effective heater actuation bandwidth of more than 100 kHz (c.f. manuscript, Figure 4a). As the heater bandwidth dictates the overall system bandwidth, these insights are critical in achieving both offset and repetition rate locks. Still, a 100 kHz locking bandwidth would be largely insufficient to phase-lock a DFB diode laser. This means that implementing any of the conventional approaches to phase stabilisation via the pump laser and the microheater would not work. However, in the case of self-injection locking (which has not been pursued so far for phase-stabilisation), the situation changes as the self-injection mechanism results in a drastic (in our case, more than 1000-fold) collapse of the linewidth of the driving laser This is now detailed in the new SI, Section 1. Through this effect, the bandwidth requirement for the actuators is significantly reduced and enables phase-locking. Moreover, leveraging synthetic reflection for self-injection locking provides a wide range of effective laser detunings for which the system can remain stable in a low-noise soliton comb state (see [1]), ensuring robust operation. To clarify the role of synthetic reflection self-injection locking, we have improved the corresponding section in the manuscript, detailing its importance for the reduction of locking bandwidth requirements and robustness of the locking operation. Overall, our results are not at all obvious and the system’s surprising simplicity, robustness, compatibility with large-scale fabrication, and ability to meet the demanding requirements of phase-coherent precision metrology represent, in our view, a major milestone for chip-integrated systems and optical precision metrology.

Below, we provide more detail and reply to the specific points raised by the Reviewer:

Details on PhCR design. Considering the only recent development of the PhCR-based systems [2–4], and self-injection locking with PhCRs [1], we have revised the manuscript in multiple points to improve clarity: First, we completed the description of the PhCR geometry in the Methods section with previously

missing information on the coupling gap. All information needed to reproduce our results is given. Second, as mentioned above, we have revised the section in the manuscript that describes the significance of the synthetic reflection by the PhCR as a way to increase the actuation range for robust phase locking and its effect of laser linewidth narrowing, which relaxes the feedback bandwidth needed for phase-locking.

Actuator tuning and physics. We thank the Reviewer for requesting a more in-depth discussion on the tuning/actuation mechanism. We agree that this would strengthen the manuscript. To address this point, we now present in the SI in Sections 1 & 2 a rigorous derivation of the tuning coefficients based on the underlying physical principles: The comb’s offset frequency is impacted predominantly by the pump diode current and, via the self-injection locking effect, by the current of the microheater which changes the resonance frequency of the microresonator. The repetition rate is impacted predominantly by the microheater through the thermorefractive effect and also through laser-resonance detuning-dependent Raman shift. The derived coefficients closely align with our experimental observations, which corroborates our understanding of the underlying physics.

Environmental sensitivity. The largest sensitivity to the environment is through temperature fluctuation. For instance, limiting resonance frequency fluctuation in the microresonator to a small (~ 10 MHz) fraction of the resonance width requires temperature stability of ~ 10 mK. This is readily achieved in our setup via standard electric cooling/heating, stabilising the temperature of both the laser chip as well as the PhCR chip within a precision of 5 mK (We have added this information to the main text for clarity). Moreover, the experimental setup is enclosed in a simple 3D-printed enclosure, insulating it from temperature fluctuations due to moving air from the lab’s air conditioning. This setup provides sufficient protection against environmental effects, and any residual temperature fluctuations are easily compensated by the phase locks acting on the microheater and laser diode current.

A non-critical limitation in our exploratory setup is the slow mechanical drift between the diode chip and the microresonator chip, which are not fixed to each other. While this enables us to easily study different laser diode/microresonator combinations, the absence of product-level photonic packaging leads to gradual misalignment between both components. In practice, this limits the operational lifetime of the soliton comb state in our setup to approximately one hour (this is not a limitation of the phase lock), largely sufficient for our proof-of-concept demonstration. If needed, mechanical misalignment can readily be mitigated through photonic packaging [5, 6], which is now routinely available through many commercial services, including temperature stabilisation of the chips, e.g. in a standard 14-pin *butterfly* package; such packaging is, however, not the focus of this present work.

Coupling losses. We thank the Reviewer for pointing out the missing information on the coupling losses between the DFB laser and the SiN chip. In the current system, using an off-the-shelf laser diode, this loss is about 3.5 dB, which falls at the better end of the 3 to 7 dB range typically reported for SIL DKS systems [5, 7, 8]. Significant improvements could be made by matching the DFB laser’s output mode profile to that of the microresonator chip’s input, which is however not the focus of the present work. A cleaved ultra-high numerical aperture optical fibre (UHNA-7) is utilised for output coupling with a coupling loss of ~ 1.7 dB. We have added an explicit mention of the coupling losses to the revised manuscript.

Long-term phase stability. A stable phase lock is maintained as long as the phase deviations between microcomb and reference remain bounded (below $\pi/2$ within the PLL) such that no optical cycles are lost. The long-term phase stability of our system can be evaluated from the Allan deviation of the in-loop and out-of-loop beat notes (c.f. manuscript, Figure 3d). To clarify the interpretation of the Allan deviation, we have completely revised the corresponding section in the manuscript, in particular with regards to interpreting the slope of the Allan deviation and the meaning of the in-loop and out-of-loop Allan deviations. If the phase deviations are bounded, the Allan deviation is expected to decrease with longer gate time τ approximately proportionally to τ^{-1} ; this is precisely what we observe. In contrast, for an imperfect phase lock with cycle slips, the phase would undergo a random walk, resulting in a $\tau^{-1/2}$ scaling of the Allan deviation (which we do not observe). Hence, the fact that the Allan deviations average approximately proportional to τ^{-1} validates the performance of the phase lock and the system’s long-term phase stability.

1. Ulanov, A. E. *et al.* Synthetic Reflection Self-Injection-Locked Microcombs. *Nature Photonics*, 1–6 (2024).

2. Yu, S.-P. *et al.* Spontaneous Pulse Formation in Edgeless Photonic Crystal Resonators. *Nature Photonics* **15**, 461–467 (2021).
3. Lucas, E., Yu, S.-P., Briles, T. C., Carlson, D. R. & Papp, S. B. Tailoring Microcombs with Inverse-Designed, Meta-Dispersion Microresonators. *Nature Photonics* **17**, 943–950 (2023).
4. Moille, G., Lu, X., Stone, J., Westly, D. & Srinivasan, K. Fourier Synthesis Dispersion Engineering of Photonic Crystal Microrings for Broadband Frequency Combs. *Communications Physics* **6**, 1–11 (2023).
5. Shen, B. *et al.* Integrated Turnkey Soliton Microcombs. *Nature* **582**, 365–369 (2020).
6. Lihachev, G. *et al.* *Frequency agile photonic integrated external cavity laser* 2023. arXiv: 2303.00425[physics].
7. Briles, T. C. *et al.* Hybrid InP and SiN Integration of an Octave-Spanning Frequency Comb. *APL Photonics* **6**, 026102 (2021).
8. Voloshin, A. S. *et al.* Dynamics of Soliton Self-Injection Locking in Optical Microresonators. *Nature Communications* **12**, 235 (2021).

Additionally, a comparative analysis with previous microcombs in terms of performance, efficiency, and cost is missing, questioning the novelty of this work.

We welcome the suggestion for a comparative analysis highlighting the novelty of our work. To address this point, we have added a comparison to previous phase-stabilised microcombs in the SI of our revised manuscript (Table 1). This comparison shows that our work represents a leap forward for microcombs, achieving full phase-stability (i.e. metrological performance) for the first time in a chip-scale system, reducing footprint/volume by three orders of magnitude, and cost by two orders of magnitude (more when produced at scale). We believe these metrics clearly illustrate the significant advancement our work represents for microcomb technology and its implications on future cross-disciplinary applications.

Given these gaps, particularly in the context of contributing new physics or significant novelty, I do not recommend publication in Nature Communications without substantial revisions to address these issues.

We thank the Reviewer for the constructive feedback and the detailed comments that have enabled us to substantially revise and improve our manuscript. We believe that this revised manuscript and the new SI now better convey the novelty of our work and the significance of this first chip-scale comb source that achieves full phase stability as needed for integrated optical clocks and timing distribution in large-scale facilities or telecommunication networks. In the revised manuscript, we also clarified the physics behind the tuning mechanism (SI, Section 1 & 2), including direct heating effects and indirect mechanisms based on the Raman effect. The insights presented in our work can guide future designs and, more generally, establish microheaters as high-bandwidth control actuators for challenging precision applications. In our view, the presented results mark a major milestone for optical precision metrology and integrated photonics, with impact on cross-disciplinary applications. Thus, we believe the results are of interest to the wide audience of Nature Communication.

In conclusion, we hope that our substantial revisions and additions satisfactorily address all comments by the Reviewer, and that the Reviewer can come to a positive recommendation of our work.

REVIEWERS' COMMENTS

Reviewer #1 (Remarks to the Author):

Since the authors took into account all my comments and made appropriate changes to the manuscript, I believe that the article can be published in its current form.

Reviewer #2 (Remarks to the Author):

I acknowledge the substantial effort and thoughtful revisions made by the authors in response to my comments and those of the other reviewers. The modifications have significantly enhanced the clarity and depth of the manuscript, addressing critical points regarding the novelty, methodology, and underlying physical mechanisms. The authors have highlighted the differentiation from previous work, the unique aspects of their phase-stabilization approach, and provided comprehensive explanations and additional data that strengthen their findings. The technical quality and overall presentation of the paper have thus improved markedly. While I recommend the paper for publication in its current form, I remain somewhat unconvinced that Nature Communications is the most suitable journal for this work. The rather specialized nature of the study may resonate better with a journal focused on photonics or precision metrology, whose community may better appreciate the topic. However, I understand that the decision ultimately rests with the editor, who determines the journal's scope and the topics it aims to cover.

Reviewer #3 (Remarks to the Author):

The revised manuscript adds a clearer overview of the PhCR design, environmental stability, and benchmarks. However, the scientific advancements presented, while notable for their integration into a chip-scale format and the incorporation of additional photonic devices, appear somewhat restricted in scope.